# Strongly-interacting bosons at 2D-1D Dimensional Crossover

Hepeng Yao[1][*], Lorenzo Pizzino[1] and Thierry Giamarchi[1]

**1** DQMP, University of Geneva, 24 Quai Ernest-Ansermet, CH-1211 Geneva, Switzerland
* Hepeng.Yao@unige.ch

February 7, 2023

## Abstract

We study a two dimensional (2D) system of interacting quantum bosons, subjected to a continuous periodic potential in one direction. The correlation of such system exhibits a dimensional crossover between a canonical 2D behavior with Berezinski-Kosterlitz-Thouless (BKT) properties and a one-dimensional (1D) behavior when the potential is large and splits the system in essentially independent tubes. The later is in the universality class of Tomonaga-Luttinger liquids (TLL). Using a continuous quantum Monte Carlo method, we investigate this dimensional crossover by computing longitudinal and transverse superfluid fraction as well as the superfluid correlation as a function of temperature, interactions and potential. Especially, we find the correlation function evolves from BKT to TLL type, with special intermediate behaviors appearing at the dimensional crossover. We discuss how the consequences of the dimensional crossover can be investigated in cold atomic gases experiments.

# 1 Introduction

Dimensionality plays an important role in the properties of quantum many-body systems, since it modifies the effects of quantum and thermal fluctuations. In three dimensions (3D), order is usually the norm at low temperatures due to moderate fluctuations and single-particle excitations generally exist. Two dimensionality (2D) usually reinforces thermal fluctuations leading to an easier destruction of perfect long-range order, replacing it by quasi-long-range order at finite temperature as in the celebrated Berezinski-Kosterlitz-Thouless (BKT) transition [1, 2]. Topological excitations can be present. In one dimension (1D), the fluctuations have even stronger effects and long-range order is usually destroyed even at zero temperature by quantum fluctuations, leading to exotic phases such as the Tomonaga-Luttinger liquid (TLL) [3, 4]. No single-particle excitations exist in this case and the behavior of the system is totally controlled by collective modes. These effects have been well explored both in condensed matter or cold atomic systems [4–7]

Although in most systems the dimensionality is well fixed, an important class of systems exists for which the dimensionality itself can be controlled either by temperature or by changing an internal parameter. This is for example the case of the organic conductors made of weakly coupled fermionic chains [8], weakly coupled spin chains and ladders [9–11], coupled bosonic chains [12–15], and fermionic or bosonic stripe phases [16–21]. This class of systems thus exhibits a dimensional crossover [22] with a drastic change of the properties and excitations when varying a parameter. Understanding such dimensional crossover is a considerable challenge with important experimental consequences. For instance, for bosonic systems, recent researches have focused on such systems in the situation of superfluid to Mott insulator transition [12, 23, 24], superfluid to normal fluid transition [25, 26], out-of-equilibrium dynamics [13, 14, 27], quantum droplets [28, 29] and supersolid phases [15].

For most condensed matter systems, a tight-binding description of weakly coupled well-defined low dimensional objects (e.g. chains) is an appropriate starting point. Using this description, mean field analysis [9, 23, 24, 53] and numerical approaches [26] have studied the effect of the quantum fluctuations of the low dimensional objects on the ordering of the system and found important differences compared to the isotropic case, both for the critical temperature and the excitation modes. For cold atomic systems however, the system is usually split into lower dimensional units by raising the periodic potential of an optical lattice [5, 13, 30–32]. Although very deep potentials lead back to a tight-binding description [5], more complex situations can occur since for intermediate potentials, the low dimensional units are not defined from the start but smoothly emerging out of the higher dimensional system. In addition, the change of the periodic potential also affects the effective interactions as well as the kinetic energy. It is thus important, especially in connection with experiments with cold atoms, to see how dimensional crossover occurs in such continuous models.

This question is particularly relevant for 2D bosonic systems which have been recently realized in a homogeneous box potential, where the BKT physics was clearly identified [6, 33–37]. Such systems can be continuously modulated to the limit of independent tubes by a unidirectional periodic potential, similarly than for 3D bosons in a trap [12]. Going continuously from the homogeneous 2D gas to the weakly coupled 1D TLL tubes offers new perspectives for the dimensional

crossover.

In the present paper, we address such a problem with a direct idea of application to realistic cold atomic systems. We study the 2D-1D dimensional crossover for a strongly-interacting continuous 2D system with unidirectional continuous lattice at finite temperature. We choose the strong interaction regime where quantum fluctuations are more pronounced. Using a quantum Monte Carlo approach, we study physical properties such as the longitudinal and transverse superfluid stiffness. We show that the one-body correlation functions evolves from BKT to TLL behavior. In addition to features in agreement with the tight-binding model, we also find additional intermediate regimes with special properties different from those of integer dimensions. We discuss the consequences of these findings for cold atomic experiments.

## 2 Model and approach

We consider a 2D cold Bose gas with repulsive two-body contact interactions subjected to the external potential $V(\mathbf{r})$, with $\mathbf{r} = (x, y)$ the position of the atom, governed by the Hamiltonian

$$\hat{H} = \sum_j \left[ -\frac{\hbar^2}{2m} \nabla_j^2 + V(\hat{\mathbf{r}}_j) \right] + \sum_{j<k} U(\hat{\mathbf{r}}_j - \hat{\mathbf{r}}_k) \tag{1}$$

where $\hat{\mathbf{r}}_j$ is the position of the $j$-th particle and $U$ a short-range repulsive two-body interaction term. We add a unidirectional lattice potential along the $y$ axis, which writes $V(\mathbf{r}) = V_y \cos^2(ky)$ with $V_y$ the potential amplitude, $k = \pi/a$ the lattice vectors and $a$ the lattice period. We use the lattice spacing $a$ and the corresponding recoil energy $E_r = \pi^2 \hbar^2 / 2ma^2$ as the space and energy units. The potential $U(\hat{\mathbf{r}}_j - \hat{\mathbf{r}}_k)$ is fully characterized by the 2D scattering length $a_{2D}$. For a 2D gas generated by a strong confinement on the transverse direction, the 2D scattering length can be expressed as a function of the 3D scattering length $a_{3D}$ and characteristic transverse length $l_\perp = \sqrt{\hbar/m\omega_\perp}$ [38,39], which writes $a_{2D} \simeq 2.092 l_\perp \exp(-\sqrt{\pi/2} l_\perp / a_{3D})$. Remarkably, the quantity $a_{3D}$ can be linked with the coupling constant $g$ both in 2D and 1D [5]. The 2D coupling constant $g_{2D}$ is [38, 39]

$$\tilde{g}_{2D} \simeq \frac{2\sqrt{2\pi}}{l_\perp/a_{3D} + 1/\sqrt{2\pi} \ln(1/\pi q^2 l_\perp^2)}, \tag{2}$$

where $\tilde{g}_{2D} = mg_{2D}/\hbar^2$ is the rescaled coupling constant and $q = \sqrt{2m|\mu|/\hbar^2}$ is the quasi-momentum. On the other hand, the 1D coupling constant $g_{1D}$ can be written as [5, 40].

$$\tilde{g}_{1D} = \frac{2a_{3D}}{l_\perp^2} \left( 1 - \frac{1.036 a_{3D}}{l_\perp} \right)^{-1}. \tag{3}$$

with $\tilde{g}_{1D} = mg_{1D}/\hbar^2$.

To study the properties of the system at finite temperature, we rely on *ab initio* quantum Monte Carlo (QMC) calculations and use path integral Monte Carlo in continuous space to simulate the Hamiltonian (1). At a given temperature $T$, 2D scattering length $a_{2D}$ and chemical potential $\mu$, we find the particle density $n$ from the counting of closed worldlines. The superfluid fraction $f_s$ in both directions is computed from the winding number estimators under periodical boundary conditions [41]. Thanks to the worm algorithm implementations [42, 43], we can compute the

one-body correlation function $g^{(1)}(x, y) = \langle \hat{\Psi}^\dagger(x, y)\hat{\Psi}(0, 0)\rangle$ in the open worldline configurations, which writes

$$g^{(1)}(x, y) = \iint \frac{dx' dy'}{L_x L_y} \langle \Psi^\dagger(x' + x, y' + y)\Psi(x', y')\rangle, \tag{4}$$

with $L_{x,y}$ the system size along the two directions. Then, the momentum distribution $D(k_x, k_y)$ can be computed from its Fourier transform and the condensed fraction $f_c^L$ is obtained from the zero-momentum portion $f_c^L = D(0, 0)/(\sum_{k_x, k_y} D(k_x, k_y))$. Due to the finite size and periodic boundary conditions, the sum is performed over $k_i = j \times 2\pi/L_i (i = x, y)$ with $j$ integers. Notably, in dimension lower than 3, there is no true condensate at finite temperature in the thermodynamic limit. The $f_c^L$ we computed here is rather a fraction of quasicondensate for a finite size system at low enough temperature, instead of the true condensate fraction in the thermodynamic limit. In practise, this is the quantity which is strongly relevant for experimental observations [44–46]. Here, we use the same QMC algorithm as Refs. [47–50]. More details about the technique is shown in Appendix. A.

## 3 Phase diagram

In Fig. 1, we show a sketch of the various regimes for strongly-interacting bosons at various temperatures and lattice depths. We focus on the temperature range $k_B T/E_r = 0.0067 - 0.2$ and the lattice potential range $V_y/E_r = 0 - 40$. Without losing generality, we consider the system size $L_x = 25a$, $L_y = 5a$, particle density $n = N/(L_x L_y) = 0.5a^{-2}$ and $a_{2D} = 0.01a$. In the strictly-2D and strictly-1D limits, these values lead to rescaled coupling constants $\tilde{g} = mg/\hbar^2$ on the scale $\tilde{g}_{2D} \simeq 1.36$ and $\tilde{g}_{1D} \simeq 10$, see detailed calculations in Appendix. A. For cold atomic gases, the criteria of strongly-interacting regime are the coupling constant $\tilde{g}_{2D} \geq 1$ in 2D and the Lieb-Liniger parameter $\gamma = \tilde{g}_{1D}/n \gg 1$ in 1D [5–7, 51]. With the particle density we have chosen, we have $\tilde{g}_{2D} \simeq 1.36 \geq 1$ and $\gamma > 10 \gg 1$. Therefore, our system remains in the strongly-interacting limit in the full range of parameters considered in Fig. 1. Moreover, for all the results we show later, we have performed the finite-size analysis and show they should hold qualitatively at different system size and anisotropy (see details in Appendix. B).

In Fig. 1, we find five different regimes based on the superfluid fraction along the two directions and the condensate fraction. At low temperature, the system is a quantum degenerate gas in different dimensionalities. When $V_y$ is small, the system is a 2D quantum gas with a weakly modulated density (2D, yellow). A larger potential $V_y/E_r \sim 7$ causes important enough modulations in the density $n_{min}/n_{max} \leq 5\%$ and the system cannot be viewed as a 2D system any more, but starts to be built of coupled lower dimensional 1D units ("tubes"). This is denoted by the shaded blue region. When the modulation becomes large enough $V_y/E_r \sim 10$, the 1D units are well formed and one can consider with a high accuracy [5] that the system is described by a tight-binding Hamiltonian

$$\hat{H} = \sum_i \left[\hat{H}_{1D,i} - t(\hat{b}_{i+1}^\dagger \hat{b}_i + \text{h.c.})\right], \tag{5}$$

with the tunneling

$$t = \frac{4}{\sqrt{\pi}} V_y^{3/4} E_r^{1/4} e^{-2\sqrt{V_y/E_r}} \tag{6}$$

and $\hat{H}_{1D,i}$ the 1D bosonic Hamiltonian. Depending on the temperature and the tunneling, the tubes can be either coherently (C-1D, dark blue) or incoherently (I-1D, light blue) coupled [23, 24, 26].

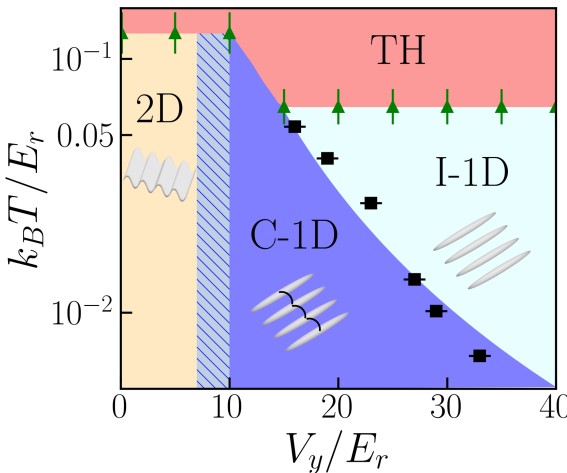

Figure 1: A schematic "phase diagram" characterizing the dimensional crossover as a function of temperature $T$ and $y$-direction lattice depth $V_y$ for strongly-interacting bosons, in units of the recoil energy $E_r$. Below quantum degeneracy, regimes are: 2D gas with modulated density (2D, yellow), well formed single mode 1D tubes coupled by tunneling coherently (C-1D, dark blue) and incoherently (I-1D, light blue). At high temperature, the system reaches a thermal phase (TH, red). The crossover region from 2D to C-1D is marked as the shaded area. All the other crossover lines are obtained from the asymptotic fits of the QMC data points (black squares and green triangles).

In I-1D regime, the tunneling is small enough that the system can be described by a purely 1D Hamiltonian. It is identified by $f_s^y = 0$ in thermodynamic limit. For our finite size system, we use the criterion $f_s^y < 0.1\%$, represented by black square points in Fig. 1. Correspondingly, the condensate fraction $f_c^L$ remains finite at this crossover and drops to a small constant in I-1D regime.

Here, we give one example of the behavior for the three quantities computed from QMC along the cut of Fig. 1 at fixed temperature $k_B T/E_r = 0.0135$, see Fig. 2.(a1-a3). The background colors indicate the regimes of the system, namely 2D (yellow), C-1D (dark blue) and I-1D (light blue). The shaded area is the crossover between 2D and C-1D regime. By increasing the potential amplitude $V_y$, we find both the $y$-direction superfluid fraction $f_s^y$ and the condensate fraction $f_c^L$ drop, while the $x$-direction superfluid fraction $f_s^x$ remains almost a constant at a large value. At the crossover between the C-1D and I-1D regime, the $y$-direction superfluid fraction $f_s^y$ drops to zero and the condensate fraction $f_c^L$ converges to a small value and stays almost like a constant. The behavior of $f_s^y$ and $f_c^L$ is similar in most of the other cuts in the low temperature regime of Fig. 1. Therefore, we can judge the crossover potential $V_{y,\text{cross}}$ for entering the I-1D regime by the condition $f_s^y < 0.1\%$. Correspondingly, we always find $f_c^L$ saturates at small constant values at the obtained $V_{y,\text{cross}}$ which further confirms the validity of this judgment. In the example we show in Fig. 2, the crossover between C-1D and I-1D regime is found at $V_y = 28.0 E_r$.

Now, we turn to the detailed discussion about the crossover line between C-1D and I-1D regimes. At large enough potential amplitude $V_y$, we can write the effective tunneling as a function of $V_y$ according to Eq. 6. Above, we have discussed that one can compute a crossover potential $V_{y,\text{cross}}$ at each given temperature $T$. Equivalently, at each given $t$ (or equivalently $V_y$), we can find a crossover temperature $T_{\text{cross}}$ above which the system enters the I-1D regime. In Fig. 2.(b), we plot the detailed data of $T_{\text{cross}}$ as a function of $t$ from the QMC calculations (blue points). Here,

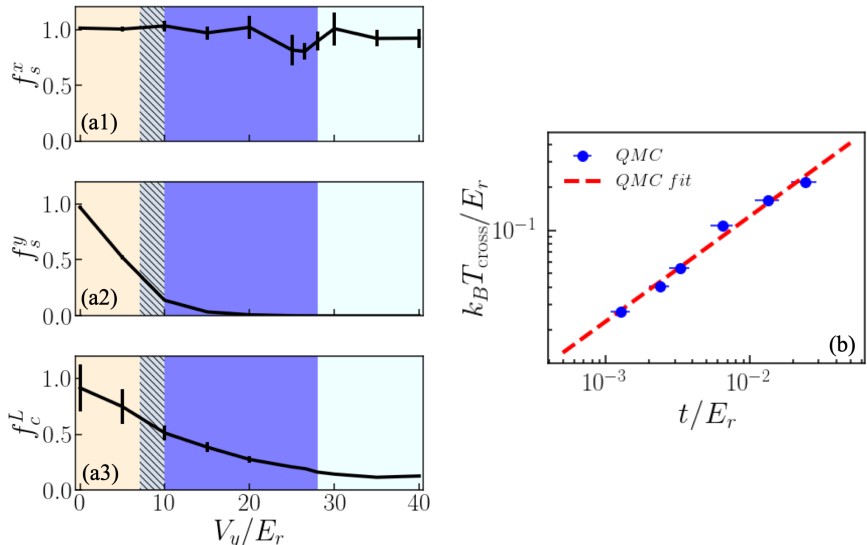

Figure 2: (a) The example of detailed data for the superfluid fraction along the two directions $f_s^x$ (a1) and $f_s^y$ (a2), and the condensate fraction $f_c^L$ (a3), as a function of the $y$ direction lattice depth $V_y$ at fixed temperature $k_B T/E_r = 0.0135$. The three regions are 2D (yellow), C-1D (dark blue) and I-1D (light blue). The shaded area is the crossover between 2D and C-1D regime. The crossover between the C-1D and I-1D regime is estimated to be at $V_y = 28.0 E_r$. (b). The crossover temperature $T_{cross}$ as a function of the transverse direction tunneling $t$. The QMC data is plotted as blue balls, and they follow a fit of the scaling $T_{cross} \sim t^\nu$ with $\nu \simeq 0.72 \pm 0.04$ which is presented in red dashed line. Here, the system size is $L_x, L_y = 25a, 5a$, the particle density is $n = 0.5 a^{-2}$, and the scattering length is $a_{2D} = 0.01a$.

the range of $t/E_r$ in Fig. 2.(b) corresponds to the lattice potential $V_y/E_r$ from 10 to 32. From the QMC data, we find the scaling $T_c \sim t^\nu$ and it shows a linear behavior in log-log scale. From the linear fit (red dashed line), we find $\nu \simeq 0.72 \pm 0.04$. Remarkably, the exponent we found here is less than 10% different from the mean field prediction $\nu_{MF} = 2K/(4K - 1) \simeq 0.67$ for the discrete model in the thermodynamic limit [24, 26], with $K$ the the Luttinger parameter [4, 7]. This indicates that our results on the continuous model are thus confirming fully the tight-binding results for large potentials $V_y/E_r \geq 10$.

Moreover, at high enough temperature, the superfluidity of the system is totally destroyed, leading to a thermal phase (TH). For the finite size we considered here, this regime is determined by $f_s^x, f_s^y < 0.1\%$, see red region in Fig. 1. Note that the critical temperatures between thermal and quantum regimes are not equal in different regimes of dimension due to the fact that the long-range correlations are much more fragile in 1D comparing with the 2D case (see detailed discussions below).

## 4   Longitudinal superfluidity

The longitudinal superfluid fraction $f_s^x$ exhibits interesting properties at the dimensional crossover. Since there is no lattice potential directly applied on this direction, the behavior of $f_s^x$ reflects

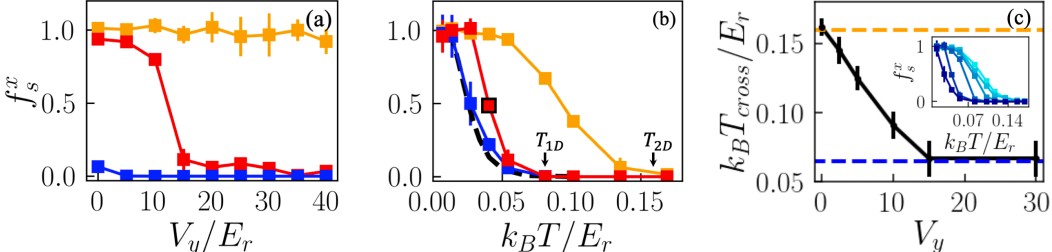

Figure 3: Superfluid fraction along $x$ direction $f_s^x$, at 2D scattering length $a_{2D} = 0.01a$, particle density $0.5a^{-2}$ and system size $L_x, L_y = 25a, 5a$. (a). The dependence of the lattice potential amplitudes $V_y$ at three different temperature $k_B T/E_r = 0.014$ (orange), 0.056 (red) and 0.143 (blue). (b). The temperature dependence at three different potential amplitudes $V_y/E_r = 0$ (orange), 15 (red) and 30 (blue). The black dashed line presents the analytical formula Eq. (7). The estimated crossover temperatures at the two extreme cases is marked as $T_{2D}$ (2D) and $T_{1D}$ (I-1D). (c). Crossover temperature to the thermal phase $T_{cross}$ as a function of the potential amplitudes $V_y$, judged from the longitudinal superfluid fraction $f_s^x$. The orange and blue dashed lines present the estimated temperatures $T_{2D}$ and $T_{1D}$. Inset: longitudinal superfluid fraction $f_s^x$ as a function of temperature $T$ for various potential $V_y/E_r = 0$, 2.5, 5, 10, 15, 30. The darkness of color increases from light to dark blue as $V_y$ increases.

directly the effect of dimensionality. Based on the QMC results, we study $f_s^x$ along several cuts in Fig. 1.

Fig. 3(a) shows $f_s^x$ as a function of the lattice amplitude $V_y$ at various fixed temperatures. At low enough temperature $k_B T/E_r = 0.014$ (orange), the long-range coherence is preserved both in 2D and 1D limit for a system of finite size. Thus, $f_s^x$ remains at a large value around 1 for any potential strength considered here. On the contrary, at high enough temperature $k_B T/E_r = 0.143$(blue), the superfluidity is completely destroyed in both 2D and 1D limits, which leads to $f_s^x \simeq 0$ in both regimes.

At intermediate temperature, the dimensional crossover is visible since 1D superfluidity is fragile to the temperature while the 2D one survives. We choose $k_B T/E_r = 0.056$ (red) as an example. At small $V_y$, the system is in 2D regime and the quantity $f_s^x$ is at a large value nearby 1 because of the quasi-long-range order in the BKT phase even at finite $T$. When increasing $V_y$, the quantity $f_s^x$ starts to drop. In the range $V/E_r = 10 - 15$, its value shows a sudden and large decrease and hits a small value around zero since correlation decreases exponentially at finite $T$ in the I-1D regime. The crossover between the C-1D and I-1D regimes happens at $V_y = 15E_r$ at this temperature (see Fig. 1). Remarkably, the $x$ direction superfluid fraction shows a dramatic change although we only increase the transverse lattice amplitude $V_y$ without changing any parameters along $x$ direction.

Fig. 3(b) shows $f_s^x$ as a function of the temperature $T$ at various lattice amplitudes $V_y$, for $V_y/E_r = 0$ (orange), 15 (red), 30 (blue). At $V_y = 0E_r$, the system is a purely-2D gas. In the thermodynamic limit, its $x$-direction superfluidity should hit zero through the BKT transition at the critical temperature $n_s\lambda_T^2 = 4$, with $\lambda_T = \sqrt{2\pi\hbar^2/mk_B T}$ the de Broglie wavelength and $n_s = nf_s$ the superfluid density [5, 6]. Here, we can estimate the $T_{2D}$ of our finite size system by taking $n_s \simeq 0.5a^{-2}$ and we find $k_B T_{2D} \simeq 0.16E_r$ which is in agreement with the QMC data. On the contrary, in the strictly-1D limit $V_y = 30E_r$, $f_s^x$ should follow the finite temperature properties of

a 1D superfluid at finite size $L_x$ [52],

$$f_s = 1 - \frac{\pi u K}{L_x k_B T} \left| \frac{\theta_3''(0, e^{-2\pi u K/L_x k_B T})}{\theta_3(0, e^{-2\pi u K/L_x k_B T})} \right| \tag{7}$$

with $u$ the sound velocity, $\theta_3(z, q)$ the Jacobi Theta function of the third kind and $\theta_j''(z, q) = \partial^2 z \theta_j(z, q)$. As seen in Fig. 3(b), there is excellent agreement between Eq. (7) (black dashed line) and the QMC data (blue squares). In the I-1D regime, the correlation function decays exponentially at distance $x > \xi$, with $\xi = 2\beta u K/\pi$ [4]. Thus, $f_s^x$ drops to almost zero when $\xi(T) \ll L_x$, defining a (size dependent) temperature $T_{1D}$. Taking $L_x/\xi \simeq 8$, we find $k_B T_{1D} \simeq 0.065 E_r$.

The intermediate case $V_y = 15 E_r$ (red squares) shows that, for small temperatures $k_B T \leq 0.03 E_r$, $f_s^x$ follows essentially the 2D curve due to the coherent tunneling in both directions. Increasing further the temperature leads to a rapid drop of $f_s^x$ signaling the dimensional crossover. At $k_B T \geq 0.06 E_r$, the particles can hardly execute any coherent tunneling between tubes and the value $f_s^x$ hits and follows the 1D curve. Around $k_B T \simeq 0.04 E_r$, the system cannot be considered either as a 1D or 2D superfluid as shown by the intermediate value of $f_s^x$ (square with black frame).

In Fig. 3(c), we further compute the crossover temperature to the thermal phase $T_{cross}$ as a function of the potential amplitudes $V_y$. This is judged by the longitudinal superfluid fraction $f_s^x$, see inset. When $V_y/E_r = 0$, the value of $T_{cross}$ is nearby the estimated $T_{2D}$. Then, $T_{cross}$ gets lower when $V_y$ increases. This is due to the fact that the anisotropy induced by the transverse lattice decreases the temperature to reach quantum degeneracy. At $V_y/E_r = 15$, it reaches the estimated $T_{1D}$ and stays. Notably, the temperature $T_{cross}$ we computed here is a crossover temperature specifically corresponded to our finite-size system. Strictly speaking, it is not the BKT temperature which should be judged from a finite-size analysis, although the value $T_{cross}(V_y = 0)$ is not far from the estimated $T_{2D}$. Nevertheless, it will be worth to investigating analytical calculations for the BKT temperature at anisotropic systems and compare it with the numerical data at finite size. Such calculations may be carried out by self-consistent harmonic approximation, see for instance Ref. [53].

## 5   Correlation functions

Let us now turn to the correlation functions $g^{(1)}(x, y)$, which measures directly the degree of coherence both along and perpendicular to the direction of the potential $V(y)$. We compute $g^{(1)}(x, y)$ at fixed temperature $k_B T/E_r = 0.021$, particle density $n = 0.5a^{-2}$, and system size $L_x, L_y = 25a, 5a$. These parameters allow us to access all the quantum degenerate regimes (see Fig. 1). To capture the continuous evolution from 2D to 1D, we take four lattice potentials $V_y = 0 E_r$ (homogeneous 2D), $5 E_r$ (strongly modulated 2D), $10 E_r$ (crossover to C-1D) and $32 E_r$ (I-1D) as examples. The results are shown in Fig. 4.

Fig. 4(a1)-(d1) show the full correlation function $g^{(1)}(x, y)$ in different regimes. Various regimes are clearly visible. For (c1)-(d1), the potential $V_y$ is large enough that we are, for the temperature considered, essentially in the tight-binding description of single mode TLL. The decay along $y$ in (c1) shows the well-defined periodicity in $y$ that one could expect for wavefunctions corresponding to the ground state of an harmonic oscillator, see also Appendix C. In this regime, one can decompose the wavefunction $\Psi(x, y)$ into the Wannier basis at large $V_y$. It writes $\Psi(x, y) = \sum_j b_j \phi(x, y - aj)$ with $\phi(x, y - aj)$ the local wavefunction on site $j$. This allow us to

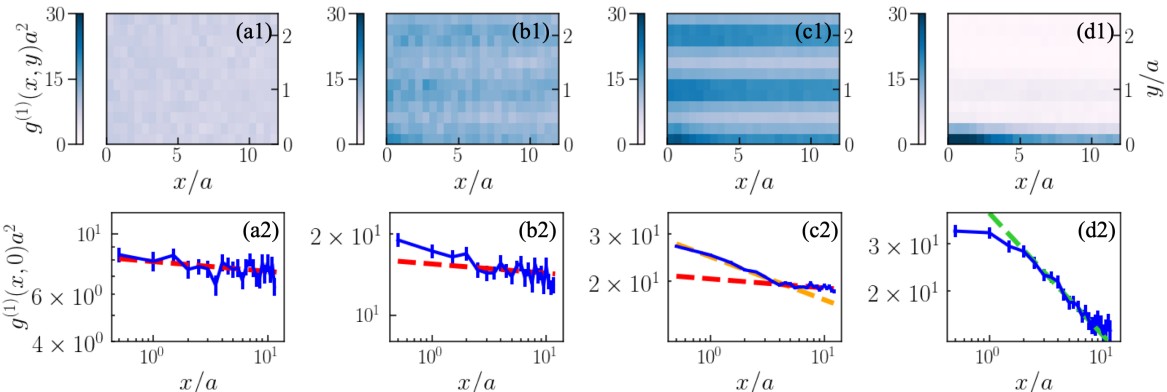

Figure 4: Correlation function of the system for 2D scattering length $a_{2D} = 0.01a$, particle density $n = 0.5a^{-2}$, temperature $k_B T/E_r = 0.021$ and system size $L_x, L_y = 25a, 5a$. Subfigures (a1)-(d1) show the full correlation function $g^{(1)}(x, y)$ for four different potential depths $V_y = 0E_r, 5E_r, 10E_r$ and $32E_r$, where the system is in homogeneous 2D, 2D with shallow lattice, crossover to C-1D, and I-1D regimes correspondingly. Subfigures (a2)-(d2) are the cuts along $x$ direction, namely $g^{(1)}(x, 0)$. The dashed lines in (a2)-(d2) are the linear fits in different regimes in the log-log scale (see details in the text).

write the correlation function $g^{(1)}(x, y)$ as

$$g^{(1)}(x, y) = \sum_j \langle \hat{b}_j^\dagger \hat{b}_0 \rangle \phi^*(x, y - aj) \phi(0, 0). \tag{8}$$

While increasing the lattice potential $V_y$, two different processes appear in Eq. (8). On one hand, the harmonic approximation becomes more accurate nearby the potential minimum and the wavefunction $\phi(x, y)$ can be better approximated by the ground state of the harmonic oscillator. This enhances the existence of the periodic pattern. On the other hand, the term $\langle \hat{b}_j^\dagger \hat{b}_0 \rangle$ evolves from an algebraic decay into an exponential decay, which weakens the periodic pattern. Thanks to the competition of these two processes, the periodic pattern evolves non-monotonically along the dimensional crossover (see Appendix. C). At the temperature chosen, (d1) shows a total loss of transverse coherence even between neighboring tubes indicating the entrance of the I-1D regime, while (c1) is at crossover to C-1D region with still excellent transverse coherence. Cases (a1)-(b1) are clearly beyond the tight-binding description, where the correlation varies very little along y direction due to the yet strong coherence in the transverse direction.

Let us now turn to the study of the $x$ direction correlation, which exhibits a stronger decay while we raise the transverse lattice potential. In Fig. 4(a2)-(d2), we plot the $x$-direction correlation $g^{(1)}(x, 0)$ in log-log scale. In Fig. 4(a2), the systems follows the property of 2D homogeneous quantum gas. It exhibits a BKT type of decay $g^{(1)}(x, 0) \sim x^{-\alpha_{2D}}$ with $\alpha_{2D} = 1/n_s \lambda_T^2$ the inverse quantum degeneracy parameter [5, 6]. By a linear fit in log-log scale(red dashed line), we find $\alpha_{fit} = 0.036 \pm 0.012$, which fits well with the expected value at the considered temperature $\alpha_{2D} = 1/n_s \lambda_T^2 = 0.032$. The extreme opposite case is the I-1D regime where the correlation function can be depicted by TLL theory at large distance. It follows $g^{(1)}(x, 0) \sim x^{-\alpha_{1D}}$ with $\alpha_{1D} = 1/2K$ linked with the Luttinger parameter $K$ [4, 7]. In the considered case, we have $K \simeq 1.01$ and the expected scaling parameter $\alpha_{1D} \simeq 0.50$. In Fig. 4(d2), we perform the fit (green dashed line)

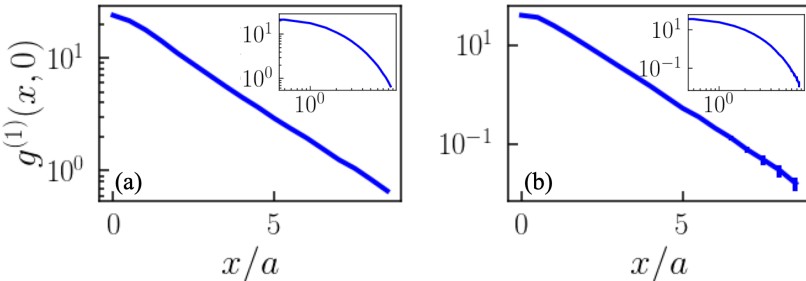

Figure 5: Longitudinal correlation function $g^{(1)}(x,0)$ for systems at high temperature $k_{\mathrm{B}}T/E_{\mathrm{r}} = 0.2$. Here, we take the 2D scattering length $a_{2\mathrm{D}} = 0.01a$ , particle density $n = 0.5a^{-2}$, and system size $L_x, L_y = 25a, 5a$. The system are in the two integer dimensionalities: (a). $V_y/E_{\mathrm{r}} = 0$, strictly 2D, (b). $V_y/E_{\mathrm{r}} = 30$, isolated 1D tubes. The main figures are in semi-log scale while the insets are in log-log scale. Both figures show an exponential decay which is expected in the thermal regime

and find $\alpha_{fit} = 0.46 \pm 0.04$ which agrees well with the expected value. Here, one should note that at a much larger distance, the $g^{(1)}(x,0)$ will decay exponentially due to the small but finite temperature [4,7]. However, it is beyond the system size we considered here.

The two intermediate cases Fig. 4(b2)-(c2) show how the longitudinal correlation evolves between these two integer dimensions. In the presence of a shallow lattice $V = 5E_{\mathrm{r}}$, a sharper drop of correlation starts to appear at short $x$ distance, while the long-range correlation remains similar to the 2D case with a similar exponent $\alpha = 0.034\pm0.013$ (red dashed line). Further increasing the potential to $V_y = 10E_{\mathrm{r}}$, the system reaches the C-1D regime and an even stronger short-distance decay is observed, see Fig. 4(c2). In log-log scale, two linear regimes with different slopes are clearly found. In both regimes, we perform the fit $g^{(1)}(x,0) \sim x^{-\alpha}$ and find $\alpha_1 = 0.16 \pm 0.01$ (orange dashed line) and $\alpha_2 = 0.034 \pm 0.02$ (red dashed line) in the small and large $x$ regions correspondingly. Recovering correlations at large distances that are similar to the 2D case (a2) can be expected since, due to the coherent tunneling in the transverse direction, the system essentially keeps its 2D character at large distances. The intermediate distance behavior is however strongly modified by the presence of the potential $V_y$. Note that the regime observed in (b2) and (c2) at short distance is *not* the short distance 1D powerlaw regime which is naturally expected in a tight-binding description. The continuous system thus offers in this intermediate coupling range of $V_y$ interesting new behaviors that will be worth investigating in more details. One possible extension is to compute the BKT prediction of the correlation function for 2D systems in the presence of the unidirectional periodic potential, and compare them with the results here.

Moreover, when the system reaches the thermal regime, one can observe an obvious change in the decay pattern of the correlation function. For both the 2D and 1D systems, the correlation function will decay exponentially above the quantum degeneracy. One example is shown in Fig. 5. Here, we compute the longitudinal correlation function $g^{(1)}(x,0)$ for system in the strictly-2D $(V_y/E_{\mathrm{r}} = 0)$ and isolated-1D $(V_y/E_{\mathrm{r}} = 30)$ regime, at high temperature $k_{\mathrm{B}}T/E_{\mathrm{r}} = 0.2$ which is above the quantum degeneracy. Clearly, we find an exponential decay in both cases, which is different from the algebraic behavior at low temperature shown in Fig. 4 (a2) and (d2).

# 6 Conclusion and experimental observability

In summary, we have studied the properties of strongly-interacting continuous bosons at 2D-1D dimensional crossover. We computed the diagram for the regimes of dimensionality at different temperatures and lattice depths. Along cuts of the diagram, we found the longitudinal superfluidity exhibits special dimensional crossover behaviors which are different from those of integer dimensions and signature the interplay of dimensionality. Further, we have studied the evolution of correlation function between the quantum degeneracy regimes of the two integer dimensions. We found the decay follows a crossover from BKT to TLL type. At the intermediate regime, we even found the short distance behavior evolving to the 1D type while the long distance behavior remains the 2D character.

The physics we describe here is adapted to current generation experiments. In cold atom experiments, low-dimensional quantum gases can be produced by loading an optical lattice potential on a continuous 3D BEC [32, 47, 54–59]. Our model gives a description that can be directly applied to such a setup. For observing our main results, the demands of experimental parameters are temperature range $k_\mathrm{B} T / E_\mathrm{r} = 0.02 - 0.1$ and interaction strength $\gamma = 20$. Such conditions can be achieved by nowadays experiments, see for instance Ref. [32]. Moreover, a box potential can cure the problem of inhomogeneity induced by a harmonic trap [33–36].

Furthermore, all the main physical quantities are detectable. By performing a time-of-flight experiment, one can measure the momentum distribution $D(k_x, k_y)$. The correlation function $g^{(1)}(x, y)$ can be directly obtained by its Fourier transform [57, 60, 61]. The strength of unidirectional superfluidity could be observed from the excitation properties along given direction [37].

# Acknowledgements

We thank Jean Dalibard and Nicolas Laflorencie for interesting discussions and comments. And we thank Christophe Berthod and Pierre Bouillot for valuable support on numerical issues. Numerical calculations make use of the ALPS scheduler library and statistical analysis tools [62–64].

**Funding information** This work is supported by the Swiss National Science Foundation under Division II.

# A Quantum Monte Carlo calculations

In most of the results of the main paper, we use the quantum Monte Carlo (QMC) calculations. More specifically, we use path integral Monte Carlo implemented with worm algorithm. Within the grand-canonical ensemble, we can compute the relevant physical quantities at a given temperature $T$, 2D scattering length $a_\mathrm{2D}$ and chemical potential $\mu$. Here, we provide more details about the QMC calculations.

## A.1 The two-body interaction

In the QMC code, the input of the interaction parameter is the 2D scattering length $a_\mathrm{2D}$. Based on this quantity, we can obtain a generalized 2D interaction propagator under pair-product approxi-

mation which can work for any interaction regime. Details about this propagator can be found in the supplementary material of Ref. [50].

In practise, the parameter $a_{2D}$ can be linked with the 3D scattering length $a_{3D}$ and coupling constant $\tilde{g}$ at different dimensions. This information has been introduced in the section "Model and approach" of the main paper. Here, we verify that our system stays in the strongly-interacting regime with the parameters we considered. In this paper, we always consider the parameters $a_{2D} = 0.01a$ and particle density $n = N/(L_x L_y) = 0.5a^{-2}$. At both two and one dimensionalities, the criteria of strongly-interacting bosons can be found in Refs. [5–7]. In the 2D case, since the condition $a/a_{2D} \gg \Lambda E_r/\mu$ with $\Lambda \simeq 2.092^2/\pi^3 \simeq 0.141$ is always satisfied, the rescaled coupling constant $\tilde{g}_{2D}$ can be estimated as $\tilde{g}_{2D} \simeq \frac{4\pi}{2\ln(a/a_{2D})} \simeq 1.36$ which satisfies the condition $\tilde{g}_{2D} \geq 1$. In the 1D case, the transverse oscillation length $l_\perp = \sqrt{\hbar/m\omega_\perp}$ is obtained from the lattice amplitude $V_y$ by $l_\perp \simeq a/\pi(E_r/V_y)^{1/4}$. Since we consider the potential range $V_y/E_r = 10 - 40$ in our paper, it leads to $l_\perp/a \simeq 0.05 - 0.1$ and we can find $\tilde{g}_{1D} = 6.0 - 10.0$ according to Eq. 3. The 1D particle density along the tube can be estimated by $n_{1D} = n_{2D} \times a = 0.5a^{-1}$. Thus, we find $\gamma > 10 \gg 1$ which satisfies the criteria of strongly-interacting regime for 1D bosons.

## A.2 Computation of observables

In the QMC calculations, the thermodynamic averages of an observable $A$ can be estimated by

$$\langle A \rangle = \frac{\text{Tr}\left[e^{-\beta(\mathcal{H}-\mu N)}A\right]}{\text{Tr}\left[e^{-\beta(\mathcal{H}-\mu N)}\right]}, \tag{9}$$

where $\mathcal{H}$ is the Hamiltonian, $N$ the number of particles operator, $\beta = 1/k_B T$ the energy scale of inverse temperature, and Tr the trace operator [41]. Thanks to the worm algorithm implementations [42,43], the configurations of worldlines can move aggressively. The particle number $N$ and density $n = N/L$ can be directly computed by the counting of worldlines. For all the numerical data of the main manuscript, we always choose the proper $\mu$ to maintain the fixed particle density $n = 0.5a^{-2}$. Also, the superfluid fraction $f_s^i = \Upsilon^i/n(i = x, y)$ can be found from the superfluid stiffness $\Upsilon^i$ along certain direction $i$, which is computed from the winding number estimator under periodical boundary condition [41]. To be more specific, it writes

$$\Upsilon_i = \frac{1}{\beta L_x L_y} \frac{m}{\hbar^2} \langle W_i^2 \rangle, \ i = x, y \tag{10}$$

with $W_i$ the winding number along $i$ direction. With the definition of Eq. (10), the value of superfluid fraction $f_s^i$ is in the range of $[0, 1]$. Moreover, in the open worldline configurations, we can compute the one-body correlation function $g^{(1)}(x, y)$ defined as

$$g^{(1)}(x, y) = \iint \frac{dx'dy'}{L_x L_y} \langle \Psi^\dagger(x' + x, y' + y)\Psi(x', y') \rangle. \tag{11}$$

This average of creation and annihilation operators can be estimated according to the worm statistics with open ends at $(x', y')$ and $(x' + x, y' + y)$ [42, 43]. Here, one should notice that for a system with sizes $(L_x, L_y)$, the correlation function is computed up to $(L_x/2, L_y/2)$. Consequently, the momentum distribution $D(k_x, k_y)$ can be obtained from its Fourier transform,

$$D(k_x, k_y) = \frac{1}{L_x L_y} \iint dx dy \ g^{(1)}(x, y)e^{i(xk_x+yk_y)} \tag{12}$$

and it is a discrete distribution with resolution $(2\pi/L_x, 2\pi/L_y)$ due to the finite size effect. Then, the condensate fraction $f_c^L$ can be obtained from the zero-momentum portion

$$f_c^L = \frac{D(0,0)}{\sum_{j,j'} D(j\frac{2\pi}{L_x}, j'\frac{2\pi}{L_y})} \tag{13}$$

with $j, j'$ taken the value of all integers. Remarkably, there should be no true condensate existed for low-dimensional bosons at finite temperature. The $f_c^L$ we computed here is a fraction of quasi-condensate generated by the finite-size effect at low temperatures. More details of the numerical techniques can be found from previous works [47–50]. Specifically, they contain the computation methods for the superfluid fraction [47, 49] and the correlation function [49], as well as the 2D scattering propagator implementations [50].

Moreover, for all the QMC results presented in the paper, we control the numerical parameter of the QMC calculation and minimize the errors induced by them. On the one hand, we use the small imaginary time step $\epsilon = 0.05 - 0.25E_r^{-1}$, which enables us to use Trotter-Suzuki approximation for estimating the short time propagator. We always make sure that the value of $\epsilon$ is much smaller than the inverse temperature $\beta$ and the corresponded standard deviation of free particle propagator $\sigma = \sqrt{\hbar^2\epsilon/m}$ is much smaller than the lattice period $a$. We have checked that our results converge with the $\epsilon$ we choose. On the other hand, we take large enough iterations to make sure the Monte Carlo statistics is sufficient. Typically, we take $N_{iter} = 3 \times 10^8$ iterations with $10^8$ warmup steps in advance. Certain parameters may demand larger values of iterations. Generally, we always make sure that a smaller $\epsilon$ or larger $N_{iter}$ will not change the physical properties presented in the main manuscript.

## B    Finite-size analysis

In the main paper, we compute the physical properties for dimensional crossover at a single system size $L_x = 25a$, $L_y = 5a$. Here, we perform the finite-size analysis and show that all the results shown in the main text remain the same qualitatively while changing the system size, as well as the anisotropy of the system. Notably, for all the discussion here, we always fix the particle density $n = 0.5a^{-2}$.

In Fig. 1 of the main paper, we produce the "phase diagram" according to the superfluid fraction $f_s$ and condensate fraction $f_c^L$. We have explained this process in Sec. 3. Especially, in Fig. 2(a1)-(a3), we give a detailed example at $k_B T/E_r = 0.0135$. Here, we reproduce this plot with five different system sizes, namely $L_x/a \times L_y/a = 15 \times 3, 20 \times 4, 25 \times 5, 35 \times 7$ and $50 \times 10$, see Fig. 6 (a1)-(a3). The darkness of the curves' colors increase with the system sizes. In Fig. 6 (a1)-(a3), we find the behaviors of both superfluid and condensate fraction remain qualitatively the same. This confirms that for any system sizes in the range of typical scales in cold atom experiments $(L_x, L_y = 10a - 100a)$, all the quantum regimes at different dimensionalities we found in Fig. 1 should remain, although the crossover point may change quantitatively. Notably, the finite value of $f_c^L$ we observed here is the fraction of quasicondensate coming from the effect of finite system size. Thus, in the thermodynamic limit $(L \to +\infty)$, no condensate can exist for a finite temperature system even at $V_y = 0$. This means the value $f_c^L$ will remain zero in the whole range of $V_y$ in that case. However, this is much beyond the scale of system size we considered here since it is at a size scale much larger than experimental conditions in cold atoms.

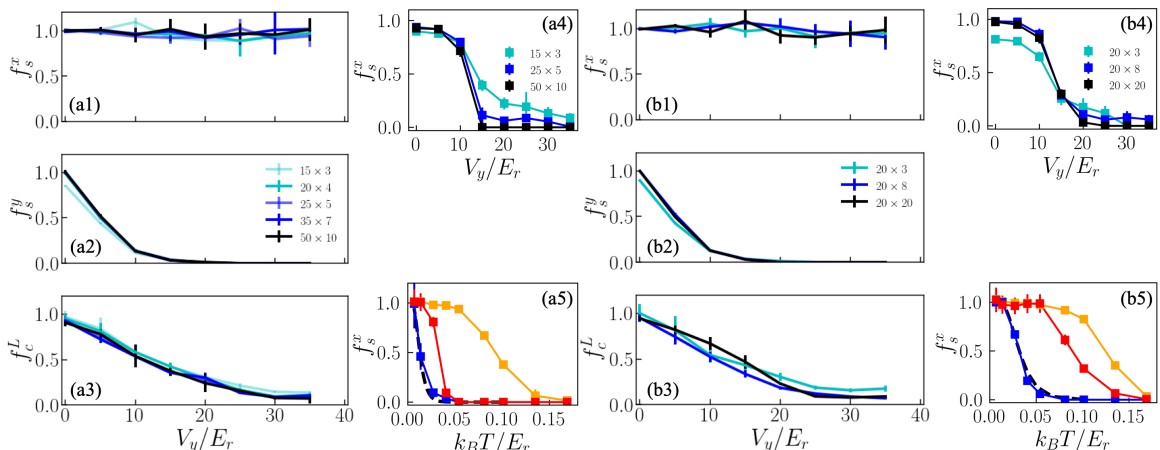

Figure 6: Finite size analysis for the superfluid and condensate fraction. The subfigures (a1)-(a3) are same plots as Fig. 2 (a1)-(a3), with system sizes $L_x/a \times L_y/a$ equal to $15 \times 3$, $20 \times 4$, $25 \times 5$, $35 \times 7$ and $50 \times 10$. Subfigure (a4) reproduces the curve at intermediate temperature $k_B T/E_r = 0.056$ in Fig. 3 (a) of the main paper, with system sizes $L_x/a \times L_y/a$ equal to $15 \times 3$, $25 \times 5$ and $50 \times 10$. Subfigure (a5) reproduces the Fig. 3 (b) of the main paper, with a larger system size $L_x/a, L_y/a = 50, 10$. The black dashed line presents the analytical formula Eq. (7). (b1)-(b5) are the plots of the same quantities as (a1)-(a5) with different choice of system sizes, where we check the effect of system's anisotropy. (b1)-(b4) take the system sizes $L_x/a \times L_y/a$ equal to $20 \times 3$, $20 \times 8$, $20 \times 20$, while (b5) takes $L_x/a, L_y/a = 20, 20$. In (a5) and (b5), different colors indicates different potential amplitudes: $V_y/E_r = 0$ (orange), 15 (red) and 30 (blue).

In Fig. 6 (a4)-(a5), we check the validity for Fig. 3 of the main paper under the effect of system size. In Fig. 3 (a), the main result is the crossover behavior on $f_s^x$ at intermediate temperature $k_B T/E_r = 0.056$. Here, we reproduce this curve with three system sizes $L_x/a \times L_y/a = 15 \times 3$, $25 \times 5$ and $50 \times 10$, see Fig. 6 (a4). By increasing the system size, we still observe the drop of $f_s^x$ as a function of $V_y$ and it even becomes sharper and sharper. This indicates the important dimensional crossover property we found in Fig. 3 (a) should qualitatively remain the same with different system sizes. Also, we check the validity of results in Fig. 3(b) by reproducing the plot at larger system size $L_x/a \times L_y/a = 50 \times 10$ (twice larger than the one we choose in the main paper along each direction), see Fig. 6 (a5). Similarly, we find the system at $V_y/E_r = 15$ (red solid line) follows the 2D curve at the beginning. Then, it starts to drop at $k_B T \simeq 0.03 E_r$ and shows a dimensional crossover signature where its value stay in between of the two integer dimension's. Then, it collapses with the 1D curve for $k_B T \geq 0.04 E_r$. This proves that our result in Fig. 3(b) should qualitatively hold for larger system sizes. And it confirms the potential observability of our results in different experimental setup with different system sizes.

In Fig. 6 (b1)-(b5), we reproduces Fig. 6 (a1)-(a5) by choosing system sizes at different anisotropy with fixed $L_x$. In Fig. 6 (b1)-(b4), we take $L_x/a \times L_y/a = 20 \times 3$, $20 \times 8$ and $20 \times 20$. And in Fig. 6 (b5), we take the value of isotropic case $L_x/a = L_y/a = 20$. All the results shown in Fig. 6 (b1)-(b5) confirm that the conclusions drawn from Fig. 1 and Fig. 3 of the main paper hold its universality qualitatively at various anisotropy of the system. However, one should notice that some of the results cannot hold if one goes to extremely anisotropic case. This is due to the fact

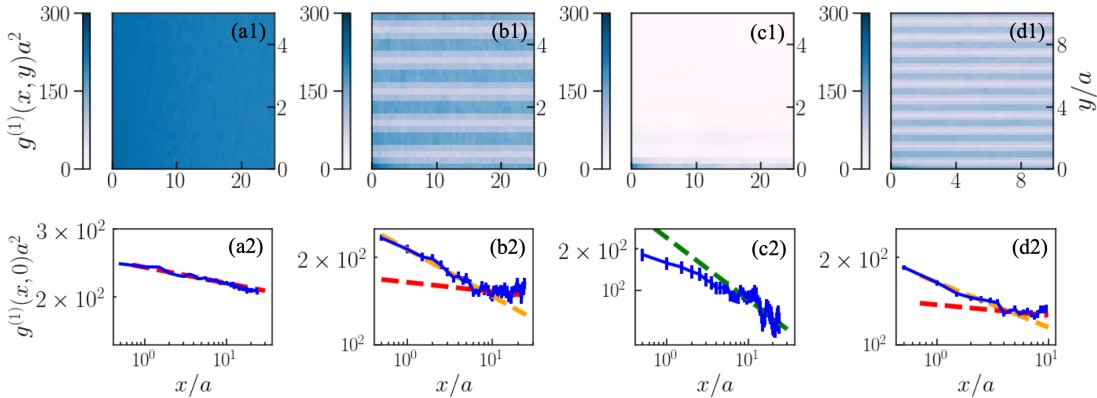

Figure 7: Correlation function of the system for 2D scattering length $a_{2D} = 0.01a$, particle density $n = 0.5a^{-2}$ and temperature $k_B T/E_r = 0.021$. Subfigures (a1)-(c1) shows the full correlation function $g^{(1)}(x, y)$ for three different potential depths $V_y = 0E_r$, $10E_r$ and $32E_r$, at system size $L_x/a, L_y/a = 50, 10$. (d1) shows $g^{(1)}(x, y)$ for $V_y = 10E_r$ and $L_x/a, L_y/a = 20, 20$. Subfigures (a2)-(d2) are the cuts along $x$ direction, namely $g^{(1)}(x, 0)$. The dashed lines in (a2)-(d2) are the linear fits in different regimes in the log-log scale (see details in the text).

that strong anisotropy will destroy the rotational invariance and the system will show a quasi-1D behavior even at $V_y = 0$. For instance, in Fig. 6 (b4), one can see such an effect starts to appear for the curve of $L_x/a \times L_y/a = 20 \times 3$, although it doesn't destroy the dimensional crossover property here. Therefore, it is important to notice that going further to extremely anisotropic case may totally undermine the physical properties discussed in our paper.

Now, we further check the properties of correlation function we found in Fig. 4 of the main paper. In Fig. 7, we reproduce the correlation function $g^{(1)}(x, y)$ at different system sizes. In Fig. 7 (a)-(c), we take the potential depths $V_y = 0E_r$, $10E_r$ and $32E_r$, at larger system size $L_x/a, L_y/a = 50, 10$ (two times larger compare with the choice of the main paper). From Fig. 7 (a1) to (c1), we plot the full correlation $g^{(1)}(x, y)$ and see clearly the non-monotonic behavior for the periodic pattern along $y$ direction. Especially, a clear and clean periodic pattern is observed in C-1D regime, see Fig. 7 (b1). Then, we plot the cut $g^{(1)}(x, 0)$ in (a2)-(c2) and find the algebraic decay $g^{(1)}(x, 0) \sim x^{-\alpha}$ in various regimes. In Fig. 7 (a2), the system is purely 2D. We find $\alpha = 0.039 \pm 0.005$ (red dashed line) which is on the scale of $\alpha_{2D} = 1/n_s\lambda_T^2 = 0.032$. In Fig. 7 (c2), the system can be treated as a purely-1D gas. We find $\alpha = 0.45 \pm 0.05$ (green dashed line) and it fits with the expected value $\alpha_{1D} = 1/2K = 0.5$ in the 1D limit. Notably, one should observe an exponential decay of $g^{(1)}(x, 0)$ at even larger system size, but this is much beyond the size we considered here. Then, we turn to the behavior of the case at dimensional crossover. In Fig. 7 (b2), we find the two slopes structure at $V = 10E_r$ still holds for the larger system size. At short distance, we find $\alpha_1 = 0.15 \pm 0.01$ (yellow dashed line) where the system is evolving into the 1D behavior. At larger distance, the system recovers the 2D behavior with a slope $\alpha_2 = 0.032 \pm 0.01$ (red dashed line). Thanks to the larger system size we take, we even find a larger region of 2D long distance behavior in Fig. 7 (b2). Moreover, we also check such special property of dimensional crossover also preserves in isotropic system. In Fig. 7 (d1), we plot $g^{(1)}(x, y)$ for system size $L_x/a = L_y/a = 20$ and lattice potential $V_y = 10E_r$. Obviously, we still find a perfect periodic pattern along $y$ direction. We further plot the cut $g^{(1)}(x, 0)$ in Fig. 7

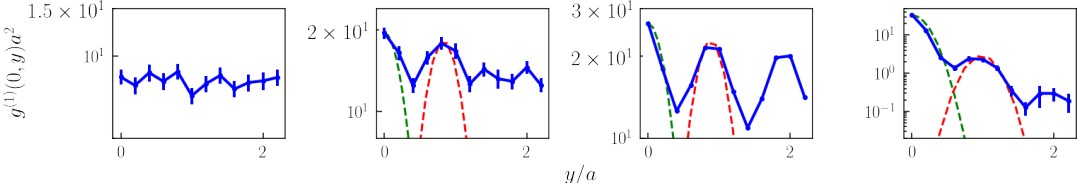

Figure 8: The transverse correlation function $g^{(1)}(0, y)$ of the system for scattering length $a_{2D} = 0.01a$, particle density $n = 0.5a^{-2}$, temperature $k_B T/E_r = 0.021$ and system size $L_x, L_y = 25a, 5a$. Subfigures (a)-(d) corresponds to four different potential depths $V_y/E_r = 0$, 5, 10 and 32. To have a better view of the periodic pattern evolution, we set the subplots (a)-(c) to have the same width along vertical axis in semi-log scale. The green and red dashed lines are the analytical formulas computed by the harmonic oscillator approximation (see details in the text).

(d2) and we still find the two slopes structure in log-log scale. We get $\alpha_1 = 0.15 \pm 0.01$ (yellow dashed line) and $\alpha_2 = 0.036 \pm 0.012$ (red dashed line), which recovers the similar dimensional crossover properties as the anisotropic case.

Above all, we conclude that all the results we shown in the main paper hold qualitatively respect to the finite-size effect in the typical scale of cold atom experiments. This universality also holds for different anisotropy of the system, as long as it is far from the quasi-1D limit.

## C Correlation function along transverse direction

In Fig. 3 of the main paper, we have shown the evolution of the full correlation function $g^{(1)}(x, y)$, where we find a non-monotonic behavior of periodic pattern along $y$ direction. Here, we give the detailed data for the cut $g^{(1)}(0, y)$ along $y$ direction with fixed $x = 0$ at the four considered cases of lattice potentials and discuss their behaviors comparing with the harmonic oscillator approximations, see Fig. 8.

In Fig. 8(a), there is no lattice potential applied and the system is a homogeneous 2D gas. Thus, the function $g^{(1)}(0, y)$ behaves similarly as the $g^{(1)}(x, 0)$, which should follow a slow decay in BKT type $g^{(1)}(0, y) \sim y^{-\alpha_{2D}}$ with $\alpha_{2D} = 1/n_s \lambda_T^2 = 0.032$. At the system size $L_y$ we consider here, it behaves almost like a constant. Increasing the lattice depth to large $V_y$, the description by Eq. 8 of the main text starts to be valid with $\phi(x, y - aj) \sim \exp[-m\omega(y - aj)^2/2\hbar]$ the ground state of quantum harmonic oscillator located at site $j$ and $\omega = k\sqrt{2V_y/m}$ the oscillating frequency. Under this harmonic oscillator assumption, we can estimate the shape of $g^{(1)}(0, y)$ nearby integer values of $y/a$. In Fig. 8(b)-(d), we plot the estimated $g^{(1)}(0, y)$ around $y = 0$ (green dashed line) and $y = a$ (red dashed line). Apparently, such a description becomes more accurate at larger $V_y$ and it leads to a stronger periodic pattern contributed from the term $\phi^*(x, y - aj)\phi(x, -aj)$ in Eq. 8. From Fig. 8(b) to (c), thanks to the fact that the term $\langle \hat{b}_j^\dagger \hat{b}_0 \rangle$ still decays slowly, we can directly observe an enhancement of the periodic pattern. Further increasing $V_y$ to the case of Fig. 8(d), although the harmonic approximation of $\phi(x, y - aj)$ becomes even more accurate, the system enters the I-1D regime by losing its coherence along $y$ direction and the strong decay of $\langle \hat{b}_j^\dagger \hat{b}_0 \rangle$ eliminates the periodic pattern. Thus, we find the periodic pattern is strongly weakened. Also, we stress that the same type of non-monotonic behavior for the periodic pattern also appears

at the other cut of $x$.

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
