# Peer review of "Strongly-interacting bosons at 2D-1D Dimensional Crossover"

_SciPost Physics_

## Round 3 · Referee Report · Anonymous (Referee 2) · 2023-2-12

Strengths

  1. Subject may be relevant to current research on cold atoms.

Weaknesses

  1. The subject is not novel and there are already numerous studies, albeit in slightly different physical settings (see, for instance, M H Fischer and M Sigrist 2010 J. Phys.: Conf. Ser. 200 012034).
  2. Conclusions are counter-intuitive and have not been convincingly demonstrated.
  3. The system studied here does not qualify as "strongly correlated".

Report

In this contribution, a relatively old subject is revisited with the use of state-of-the-art numerical methodology, and statements are made that, in my view, are not established sufficiently convincingly and definitively.

The computational methodology adopted by the authors may not be the most suitable. In spite of the title of the manuscript, the system studied here cannot be considered as "strongly interacting"; rather, it is a realization of a two-dimensional, weakly interacting dilute Bose gas in the presence of an external potential, for which mean-field type approaches might give reliable predictions without the need to carry out finite-size extrapolations that are ostensibly missing in this numerical work.

The authors claim to be observing a physical crossover from two to one dimension for a finite value of the amplitude of an external modulating potential that is only active along one direction. This is a counterintuitive conclusion, as one would expect that in the presence of tunneling the system would remain two-dimensional, possibly with a strongly anisotropic superfluid response and/or with an exponentially decreasing transition temperature, but that the system would become truly one-dimensional for a finite modulation is a puzzling contention that requires robust numerical evidence in order to be credited. Unfortunately, that kind of evidence is missing in this work.

The authors appear to be basing their conclusion that the system becomes one-dimensional on the mere observation that, at some low temperature (the lowest they can study?) and for a rather small system, the superfluid response in the transverse direction becomes "too small". This criterion is too simple – this conclusion should be supported by rigorous finite-size scaling analysis of the results, both for the superfluid fraction as well as for the correlation functions, for which a change from BKT to Luttinger-type behavior should be observed.

Unfortunately, the authors have not performed a suitably rigorous finite-size scaling analysis. The presented "finite-size scaling analysis" in the supplementary material section uses system sizes that are not sufficiently different, giving little confidence in the predictions inferred from them; nor have the scaling behavior of the superfluid density and of the correlation functions been explored. For the most part, the authors show results for a single system size (and some of the data that they show are numerically very noisy -- better statistics should be collected). It is impossible to draw the physical conclusions that they draw based on the numerical evidence that they present.

Because the numerical evidence in support of the claim made in the article is inadequate, I cannot recommend the publication of this article in any form.

Requested changes

The authors should carry out an extensive finite-size analysis of their results, by obtaining estimates on systems of significantly different sizes and showing that they conform to the physical behavior that they claim to be occurring. Particular emphasis should be given to the superfluid fraction as a function temperature as well as for the pair correlation functions, for which two distinct regimes should be convincingly demonstrated.

  • validity: low
  • significance: low
  • originality: low
  • clarity: good
  • formatting: good
  • grammar: good

Author:  Hepeng Yao  on 2023-03-03  [id 3432]

(in reply to Report 2 on 2023-02-12)

We thank the Referee for his/her work on the report. However, we respectfully disagree with some of the points of the Referee and we provide a point-to-point answers below.

Referee’s report: “The computational methodology adopted by the authors may not be the most suitable. In spite of the title of the manuscript, the system studied here cannot be considered as "strongly interacting"; rather, it is a realization of a two-dimensional, weakly interacting dilute Bose gas in the presence of an external potential, for which mean-field type approaches might give reliable predictions without the need to carry out finite-size extrapolations that are ostensibly missing in this numerical work.”

Answer: The system we considered is perfectly in the strongly-interacting regime in 2D. Here, we follow the criteria given in the review paper “Z. Hadzibabic and J. Dalibard, La Rivista del Nuovo Cimento 34(6), 389 (2011) ”. The criteria is “\tilde{g}_2D>1” because in 2D the coupling constant has a logarithmic dependence on the scattering length “g\sim log(a_sc)”. Therefore, “\tilde{g}_2D>1” will leads to “a_sc>>1”. A more detailed discussion about interaction regimes in 2D can be found also in the supplemental material of Phys. Rev. Lett.126(11), 110401 (2021). This criterion is also widely used in cold atom experiments, see for instance Phys. Rev. Lett. 110(14), 145302 (2013). In both of these references, strongly interacting properties beyond mean-field prediction are found.

Let us note that we have already presented the clarification of strongly-interacting regime in the Appendix A1 of the current manuscript, together with the references of the relevant review papers. Given the fact that the referee 3 is questioning this point again, in the process of resubmission, we may consider moving them as well as the explanation into the main text of the manuscript.

Referee’s report: “The authors claim to be observing a physical crossover from two to one dimension for a finite value of the amplitude of an external modulating potential that is only active along one direction. This is a counterintuitive conclusion, as one would expect that in the presence of tunneling the system would remain two-dimensional, possibly with a strongly anisotropic superfluid response and/or with an exponentially decreasing transition temperature, but that the system would become truly one-dimensional for a finite modulation is a puzzling contention that requires robust numerical evidence in order to be credited. Unfortunately, that kind of evidence is missing in this work.”

Answer: We do not understand the objection of the referee since this is exactly what is already claimed in the paper: in the case of zero temperature, indeed the system will always remain two-dimensional. This is what is described in Fig. 1. Since we find a scaling T_{cross}~ t^{\nu}, one needs t=0 (V_y=infinity) to reach the strictly 1D regime at T=0. However, with finite temperature, the system can be incoherently coupled due to the interplay of temperature, tunneling and interactions. Note that the fact that for very weak tunneling and finite temperature one can have a 1D regime is widely recognized in the cold atom community. This is in fact how 1D systems are created in practice in most of the cold atom experiments, see for instance the experiments of H.-C. Nägerl (Phys. Rev. Lett.115, 085301 (2015)) , G. Modugno (Phys. Rev. Lett.113, 095301 (2014) ), and I. Bloch (Nature 429, 277 (2004)).

Given the question of the referee, in next round of resubmission, we will add some additional statements to clarify this point.

Referee’s report: “The authors appear to be basing their conclusion that the system becomes one-dimensional on the mere observation that, at some low temperature (the lowest they can study?) and for a rather small system, the superfluid response in the transverse direction becomes "too small". This criterion is too simple – this conclusion should be supported by rigorous finite-size scaling analysis of the results, both for the superfluid fraction as well as for the correlation functions, for which a change from BKT to Luttinger-type behavior should be observed. Unfortunately, the authors have not performed a suitably rigorous finite-size scaling analysis. The presented "finite-size scaling analysis" in the supplementary material section uses system sizes that are not sufficiently different, giving little confidence in the predictions inferred from them; nor have the scaling behavior of the superfluid density and of the correlation functions been explored. For the most part, the authors show results for a single system size (and some of the data that they show are numerically very noisy -- better statistics should be collected). It is impossible to draw the physical conclusions that they draw based on the numerical evidence that they present.”

Answer: In this paper, we consider strongly interacting bosons in a continuous periodic potential. This is different from most previous works where a tight-binding discrete model is considered. Our model is more adapted to the situation of cold atom experiments and can capture more physics than the tight-binding case (results for the regime 0<V/E_r<10). However, due to the fact that a continuous lattice potential is studied, our calculations are limited to system sizes smaller than previous works on tight-binding discrete models and 2D homogenous systems. A continuous potential can cause a very small resolution in the position space as well as very small imaginary time step in the PIMC algorithm, which makes the calculation limited to the current system sizes we considered here. Even if so, we argue that the system sizes we considered is large enough since they are at the typical size scale in actual cold atom experiments (10a - 100a), see for instance the experimental setup of H.-C. Nägerl (Phys. Rev. Lett.115, 085301 (2015)), G. Modugno (Phys. Rev. Lett.113, 095301 (2014) ), and I. Bloch (Nature 429, 277 (2004)). Therefore, we argue that the results we present at finite sizes here are valuable for both theoretical and experimental aspects.

Moreover, although there are non-ignorable error bars on the correlation function data (which is normal due to the numerical difficulty we mentioned above), thanks to the strong interaction we considered, the decay exponent in 2D (\alpha_2D~ 0.035) is different enough from the one in 1D (\alpha_1D ~ 0.5), which makes our current data sufficient enough for distinguishing the two different characters.

Referee’s report: “The subject is not novel and there are already numerous studies, albeit in slightly different physical settings (see, for instance, M H Fischer and M Sigrist 2010 J. Phys.: Conf. Ser. 200 012034).”

Answer: We certainly agree that the subject of dimensional crossover itself is not new. One of the authors of the present paper has made several contributions in this respect even well before the reference mentioned by the referee (see e.g. TG, Theoretical Framework for Quasi-One Dimensional Systems’’, Chem. Rev. 2004, 104, 11, 5037–5056 (2004), TG.Quantum Phase Transitions in quasi-one dimensional systems’’ https://arxiv.org/pdf/1007.1029.pdf (chapter of the book Understanding Quantum Phase Transitions, ed. Lincoln D. Carr (CRC Press / Taylor&Francis, 2010) and references therein for spins, bosons and fermions, and of course the references [23,24] for example of the current paper for bosons). There are relevant studies in the tight-binding limit of our physical systems, which we already cite in the introduction (Refs. [9,23,24,26,53]). However, in all these references (including the one mentioned by the Referee), a discrete tight-binding model is considered and used with a mean-field approach. In our paper, there are two main differences: 1, We consider a continuous lattice potential which is more adapted to experimental system in cold atoms, and we perform a study beyond mean-field. 2. The fact that we consider strongly-interaction regimes will also lead to the failure of mean-field theory. In the case we considered, we can capture the full dimensional crossover for V/Er evolving continuously from 0 to 30, and observe results which cannot be described by tight-binding discrete models. Therefore, we argue that the results of this paper are new, in some regime complementary of the ones in the previous studies. For the motivation of the work as well as its comparison with previous works on dimensional crossover, we believed that we had made a clear statement in the introduction of the current manuscript. Note that the other Referees do not object on this point. In the first round of review, Referee 1 says “the results are sound, with direct implication to experiments” and Referee 2 (comment) says our work “presents various interesting results, which characterize the emerging dimensional crossover. In view of possible experimental realizations of such a dimensional crossover in the realm of atomic quantum gases, these theoretical findings are important. 

Nevertheless given the question of Referee 3, we will consider in the process of resubmission adding additional sentences on that point in the introduction.

---

## Round 3 · Referee Report · Anonymous (Referee 1) · 2023-3-24

Report

The authors make a clear statement on the validity of the results in the revised manuscript, which is important given small and anisotropic system sizes considered. As the authors claim the results are new and will indeed have a significant impact on future works on dimensional crossovers with strongly interacting bosons in continuous periodic potentials. I find the manuscript fit for the journal and recommend its publication.

---

## Round 3 · Referee Report · Anonymous (Referee 3) · 2023-3-30

Strengths

1- Numerical study of 2D-1D dimensional crossover 2-Deals with experimentally realistic set-up

Weaknesses

1- Anisotropic superfluid density and its implications for the BKT transition 2- Clarifications of various aspects

Report

In their response from 07.02.2023 the authors have not properly answered my comment from 18.01.2023 concerning the anisotropy of the superfluidity density and its impact upon the BKT transition. In the literature it is established that in an anisotropic situation in 2D the proper BKT transition criterion is given by the geometric mean of the two superfluid densities, see for instance PRL 113, 165304 (2014) and the references therein. Thus, to properly determine the BKT transition in the anisotropic system at hand, boils down to determine numerically both superfluid densities. A separate RG calculation, which would indeed be a an independent project, is not necessary. I suggest that the authors look into this during the resubmission process and improve the manuscript accordingly. This would give the authors also the possibility to take into account the various suggestions of the other referees. In particular, it seems to be appropriate to clarify more precisely

1) why the system studied here can be considered as "strongly interacting"

2) to which extend one reaches a 1D regime or, a wording which I would prefer more, a quasi-1D regime

3) why the considered system sizes are considered to be sufficient from an experimental point of view, although from a theoretical point of view a more extensive finite-size scaling would have been preferable

4) the literature of dimensional crossover by comparing, for instance, the 1D-2D crossover also with the 1D-3D crossover: Phys. Rev. Lett. 113, 215301 (2014), Phys. Rev. Lett. 117, 235301 (2016), Phys. Rev. Lett. 130, 123401 (2023)

Requested changes

see above

---

## Round 3 · Author Response

Dear editor,

We thank you for transmitting the Referees’ reports. We thank the Referees for their positive assessment of our work and the constructive comments/suggestions they made. Here, we submit a revised version of the manuscript by addressing all the comments/suggestions. Below, we also provide our answers to the Referees' comments and indicated the corresponding changes. We hope that with the changes, the paper is now suitable for publication in SciPost Physics.

Sincerely yours,
Hepeng Yao, Lorenzo Pizzino and Thierry Giamarchi

Answer to Referee 1:

Referee's comment/question 1 :"Why is there no complex conjugate term present in Eq. 5 for the tunneling part (Eq. 6)?"

Answer: We thank the Referee for pointing out this mistake. We have added the complex conjugate in Eq. 5.

Referee's comment/question 2 :"In the phase diagram (Fig. 1), the thermal regime is identified by the superfluid fractions being below 0.1 %. I am wondering if this can be corroborated by the decay of the correlation function (which in 2D changes from power-law to exponential in the thermal regime)? One might also expect a characteristic change in 1D?"

Answer: We thank the Referee for this interesting question. Indeed, the transition to thermal regime can be corroborated by the decay of correlation function, both in 2D and 1D regimes. In both cases, one expects an exponential decay in the thermal regime, which is totally different from the power-law decay observed at low temperature as in Fig. 4. To better explain this point, we run calculations at high temperature for V_y/E_r=0 and 30, namely the strictly-2D and isolated-1D regime. The calculations are performed at the temperature k_B T/E_r=0.2 where the system reaches thermal phase in both dimensionalities, see Fig. 5 of the revised manuscript. In both cases, we see a clear exponential decay which is different from the power-law decay observed in Fig. 4. It confirms the statement that the transition to thermal regime can be corroborated by the decay of the correlation function.

In the revised manuscript, we have added the discussion on this point in the last paragraph of Section 5, together with the numerical data in Fig. 5.

Answer to Referee's minor remarks: We thank the Referee for his/her careful reading and pointing out the minor remarks. We agree with all of them and we have addressed all of them carefully.

Answer to Referee 2:

Referee's comment/question 1 :"On page 7 the BKT critical temperature at V_y = 0 Er is compared with the quantum Monte Carlos results, which corresponds to the 2D regime. Here the question arises which BKT critical temperature emerges at a finite potential depth V_y, when the superfluidity becomes anisotropic. And how this BKT critical temperature compares with the results from the quantum Monte Carlo simulations."

Answer: We thank the Referee for this interesting question. From the numerical aspect, now we perform the quantum Monte Carlo calculations for f_s^x as a function of T at various values of V_y and compute the crossover temperature to the thermal regime T_{cross}. Then, we plot T_{cross} as a function of V_y in Fig. 3(c) of the revised manuscript. At V_y=0 E_r, the value of T_{cross} fits with T_{2D} which is the estimated BKT critical temperature in strictly-2D regime. Then, increasing V_y leads to decrease the value of T_{cross}. This suggests that the anisotropy induced by the unidirectional periodic lattice makes it harder for the system to reach quantum degeneracy. Further increase V_y up to 15 E_r, T_{cross} reaches the value T_{1D} which is estimated by the 1D Tomonaga-Luttinger theory.

From the analytical aspect, it will indeed be interesting to compare with the BKT temperature in anisotropic system. One possible solution is to perform self-consistent harmonic approximation (SCHA). Such calculation has been performed in 2D anisotropic XY model, see “J.-S. You et al Phys. Rev. A 86, 043612 (2012)” (Ref. [30] of the revised manuscript). The SCHA calculation shows that T_{BKT} decreases with the increase of anisotropy, which fits with the QMC results qualitatively. However, the value of T_c from the two calculations doesn’t match quantitatively. The results differ less in the limit of very anisotropic system and isotropic case. Notably, Ref. [30] considers the discrete case which is different from the continuous periodic potential we treat in the paper. More quantitative results adapted to our model, to the best of our knowledge, are still lacking. To find T_{BKT} at finite V_y in our model deserves a thorough calculation with methods like SCHA or renormalization group, which we feel is beyond the scope of the current manuscript and deserves to be an independent project. However, we agree that this is an interesting outlook based on the results in our paper and should definitely be mentioned in the manuscript.

We have added the numerical data in Fig. 3(c) and corresponding statement in the last paragraph of section 4 “longitudinal superfluidity”. We also added in the text: “It will be worth to investigating analytical calculations for the BKT temperature at anisotropic systems and compare it with the numerical data at finite size. Such calculations may be carried out by self-consistent harmonic approximation, see for instance Ref. [30]”.

Referee's comment/question 2 :"Similar questions arise for Fig. 4(b2) and 4(c2). Does the power-law obtained from quantum Monte Carlos agree with the BKT predictions for the anisotropic superfluid regime?"

Answer: We thank the Referee for this interesting question. From the numerical point of view, the results in Fig. (b2) and (c2) suggest that the long-range behavior still maintains its BKT properties with a decay exponent similarly as the strictly-2D case, while the short-range behavior shows a clearly larger decay exponent due to the anisotropy induced by the periodic potential. Considering the analytical BKT predictions, we find such results to our specific model is also lacking in the known references. Although there are known calculations for 2D XY model such as Ref. [30], they cannot describe completely our current system due to the continuous and shallow periodic potential. As for the previous question, we agree that this is an interesting outlook to our paper and we have mentioned this in the revised manuscript.

In the revised version of our paper, we add in the second last paragraph of section 5 “correlation function” the sentence “One possible extension is to compute the BKT prediction of the correlation function for 2D systems in the presence of the unidirectional periodic potential, and compare them with the results here.”, in order to point out this interesting outlook based on our current results.

Answer to Referee's minor remarks: We thank the Referee for his/her careful reading and pointing out the minor remarks. We agree with all of them and we have addressed all of them carefully.

---

## Round 3 · List of Changes

• New figures Fig. 3(c) and Fig. 5 added
  • In the section 4, we add a new paragraph in the end
  • In the section 5, we add sentences in the last two paragraphs
  • Eq. (5) are corrected
  • Typos corrected, all the minor comments of the Referees are addressed

---

## Editorial Decision

resubmitted